# Drug Use in Denmark for Drugs Having Pharmacogenomics (PGx) Based Dosing Guidelines from CPIC or DPWG for CYP2D6 and CYP2C19 Drug–Gene Pairs: Perspectives for Introducing PGx Test to Polypharmacy Patients

**DOI:** 10.3390/jpm10010003

**Published:** 2020-01-16

**Authors:** Niels Westergaard, Regitze Søgaard Nielsen, Steffen Jørgensen, Charlotte Vermehren

**Affiliations:** 1Centre for Engineering and Science, Department of Biomedical Laboratory Science, University College Absalon, Parkvej 190, 4700 Naestved, Denmark; stjo@pha.dk; 2Department of Clinical Pharmacology, Bispebjerg Hospital, University of Copenhagen, Bispebjerg Bakke 23, 2400 Copenhagen, Denmark; regitze.sn@gmail.com (R.S.N.); charlotte.vermehren@regionh.dk (C.V.); 3Department of Pharmacy, Section for Social and Clinical Pharmacy, Faculty of Health and Medical Sciences, University of Copenhagen, Universitetsparken 2, 2100 Copenhagen, Denmark

**Keywords:** drug consumption, pharmacogenomics, cytochrome P450, polypharmacy, pharmacogenomics testing, drug-drug interactions, drug–gene interaction

## Abstract

Background: The cytochrome P450 drug metabolizing enzymes CYP2D6 and CYP2C19 are the major targets for pharmacogenomics (PGx) testing and determining for drug response. Clinical dosing guidelines for specific drug-gene interactions (DGI) are publicly available through PharmGKB. The aim of this register study was to map the use of drugs in Denmark for drugs having actionable dosing guidelines (AG) i.e., dosing recommendations different from standard dosing for CYP2D6 or CYP2C19 DGI in terms of consumption. Methods: The Danish Register of Medicinal Product Statistics was the source to retrieve consumption in Defined Daily Dose (DDD) i.e., the assumed average maintenance dose per day for a drug used for its main indication in adults and number of users (2017 data). Clinical dosing guidelines were available from the PharmGKB website. Results: Forty-nine drugs have guidelines corresponding to 14.5% of total sales in DDD. Twenty-eight drugs have AG corresponding to 375.2 million DDD. Pantoprazole, lansoprazole, omeprazole, clopidogrel, and metoprolol constituted fifty-eight percent of the consumption in DDD of drugs having AG. The consumption of antidepressant drugs, opioids, and antipsychotic drugs were 157.0 million DDD; with 441,850 users, 48.9 million DDD; with 427,765 users, and 23.7 million DDD; with 128,935 users, respectively. Age distributions of consumption of drugs and drug combinations, e.g., for sertraline redeemed either alone or in combination with metoprolol and tramadol, are presented. Conclusion: This exploratory register study clearly showed that a large fraction of the Danish population, especially the elderly, are exposed to drugs or drug combinations for which there exist AG related to PGx of CYP2D6 or CYP2C19.

## 1. Introduction

Cytochromes P450 (CYP450) drug metabolizing enzymes are the major enzymes in catalyzing the oxidative biotransformation of 70%–80% of all drugs in clinical use to either inactive metabolites or active substances [1,2]. The polymorphism of genes encoding the CYP450 family of enzymes, and in particular CYP2D6 and CYP2C19, has attracted considerable attention as the major targets for pharmacogenomics (PGx) testing since they are highly polymorphic and thereby determining for drug response and adverse drug reactions (ADR) [3,4,5]. The Clinical Pharmacogenetics Implementation Consortium (CPIC) [6] and the Dutch Pharmacogenetics Working Group (DPWG) [7,8] both provide widely recognized clinical dosing guidelines for specific drug-gene interactions (DGI) [9,10]. These are compiled and publically available through the Pharmacogenomics Knowledgebase (PharmGKB; https://www.pharmgkb.org) [11]. Based on drug-gene scores for metabolic activity [12,13,14] DGI are classified into five distinct phenotypes defined as; “poor metabolizers” (PM), “intermediate metabolizers” (IM), “extensive metabolizers” (EM; normal activity) and “rapid and ultra-rapid metabolizers” (RM and UM) with UM having faster metabolic activity than RM. We will use the term RM covering both RM and UM throughout this manuscript. The guidelines provide, based on phenotype score, clinical recommendations such as dose adjustment, dose monitoring or avoidance of the given drugs. The FDA also recognizes the importance of DGI and has annotated a large number of drugs with considerations and actions to be taken from a PGx perspective [11].

The term phenoconversion introduces a complicating factor, which potentially can give rise to “genotype-phenotype” mismatches; a person scored as an EM or RM can be phenoconverted to a PM by co-medications [15] (drug–drug interactions). This means that the “true” number of PMs could be significantly higher compared to the number of PMs measured by PGx-testing alone. This term also refers to drug–drug–gene interactions (DDGI) [16]. That phenoconversion could alter a person’s drug metabolizing status has been shown in polypharmacy patients [17,18,19] and a recent comprehensive review underscores the importance of assessing and accounting for DGI and DDGI [4]. The guidelines provided by the PharmGKB webpage does not incorporate drug-drug interactions (DDI/DDGI) in the assessment of dose adjustments. However, the issue is recognized and initiatives have been taken to incorporate DDI/DGI in clinical decision tools e.g., youScript^®^ [19,20] which integrates PGx testing with comprehensive drug–gene and drug–drug interaction information to assess the cumulative impact of a patient’s genetics and drug regimen, and their risk for adverse drug events. In spite of many initiatives and advances in PGx implementation, significant barriers remain to use PGx-tests proactively; this includes improvement of physician’s and pharmacist´s awareness and understanding about PGx as well as convincing evidence presenting the collective clinical utility of a panel of PGx-markers in medication optimization [21]. A Health Technology Assessment report published in 2012 by the Danish Health Authority focusing on the potential use of CYP2D6 and CYP2C19 genotyping as a tool to improve antipsychotic drug treatment concluded that genotyping has the potential. However, the significant organizational hurdles and lack of evidence of PGx-tests utility as a tool for improving the antipsychotic treatment were barriers for daily routine use [22,23]. 

The aim of this register study was to map the use of drugs in Denmark by applying Anatomical Therapeutic Chemical (ATC) codes [24], for drugs having dosing guidelines (CPIC or DPWG) for CYP2D6 and/or CYP2C19, in terms of consumption of Defined Daily Dose (DDD) [24]. I.e., the assumed average maintenance dose per day for a drug used for its main indication in adults, and number of users who redeemed the drugs during 2017. In addition, consumption of inappropriate drug combinations, according to Medscape^®^ drug interaction checker [25], redeemed to users on the same day or during 2017 are given and discussed in relation to DDI/DDGI. We hope by this study to show the widespread use of drugs having PGx-based clinical dosing guidelines in order to identify areas where PGx testing could be a supporting tool in clinical decision making such as polypharmacy patients. This study present data retrieved from The Danish Register of Medicinal Product Statistics [26].

## 2. Materials and Methods

### 2.1. Clinical Dosing Guidelines

The Clinical Pharmacogenetics Implementation Consortium (CPIC) and the Dutch Pharmacogenetics Working Group (DPWG) clinical dosing guidelines for specific gene-drug pairs were used as the source. The guidelines are available through the publicly available PharmGKB homepage (https://www.pharmgkb.org/). By using the PharmGKB website all drugs redeemed in Denmark during 2017, having a CPIC or DPWG guideline for CYP2D6 and/or CYP2C19 drug-gene pairs, were mapped according to their ATC codes [24]. Drugs with guidelines were divided into drugs having an actionable guideline (AG) defined as at least one clinical recommendation (i.e., dose adjustment, dose monitoring, or avoidance of the given drug) different from the EM (normal situation) of any of the phenotypes PM, IM, or RM. Drugs having non-actionable guidelines (N-AG) were defined as drugs with no clinical recommendation different from EM of any of the phenotypes based on current clinical knowledge. N-AG are only provided by DPWG. In depth analyses of consumption were applied within ATC codes N02A (opioids), N05A (antipsychotics), and N06A (antidepressants) due to the relative large number of guidelines existing within these therapeutic groups. Assignment of FDA annotations were gathered from the FDA’s “Table of Pharmacogenomic Biomarkers in Drug Labels” for CYP2D6 and CYP2C19 [11,27]. “Actionable PGx” contains information about changes in efficacy, dosage, metabolism, or toxicity due DGI (e.g., “poor metabolizers”) relating to these two CYP-enzymes.

### 2.2. Register Data

The Danish Register of Medicinal Product Statistics [26], that comprises records of all prescriptions redeemed since 1st of January 1996, were used to retrieve drug consumption in 2017 by using Medstat.dk [26]. Consumption, identifiable with a person, is expressed as defined daily dose (DDD) [24] and number of users who redeemed prescriptions of the drugs searched for as shown in Table 1, Table 2 and Table 3 and Figure 1. Over the counter (OTC) consumption is not identifiable with a person and therfore is not part of this study. Data given for drug combinations redeemed to the same person, as shown in Table 4 and Table 5, were retrieved by using the personal identification number [28] (the CPR number), a unique identifier to all Danish inhabitants since 1968, to link the number of users who redeemed the combination of drugs shown on the same day or during 2017. In addition, Table 5 also present data on age distribution of comsumption. These data are not publically available and were provided from the same register as above by Statistics Denmark [29] upon request. This also explains the minor differences seen in numbers in Table 4 compared to Table 1, Table 2 and Table 3. Data provided by Statistic Denmark were retrieved at another time compared to Table 1, Table 2 and Table 3 and thereby small adjustments of data have occurred. For conversion to prevalence (users/1000 inhabitants), the total Danish population in 2017 was 5.748.769 (https://www.statistikbanken.dk/statbank5a/default.asp?w=1920). Drug-drug interactions were scored in severity by using Medscape^®^ drug interaction checker [25] and warnings related to CYP2D6 and/or CYP2C19 are displayed as “monitor closely” or “serious use alternate”. The dosing information, length of treatment and indication for prescribing were not recorded and ethics approval was not applicable according to Danish law, since the use of anonymized healthcare data for pharmacoepidemiological research does not require subject consent or approval from the Ethics Committee.

## 3. Results

In 2017 in Denmark 49 drugs corresponding to 469.1 million DDD had CPIC/DPWG guidelines (both AG and N-AG) for CYP2D6 and/or CYP2C19 drug gene pair’s equivalent to 14.5% of the total sale in DDD. Of these, 28 drugs have AG corresponding to 375.2 million DDD i.e., 80.0% of total consumption in DDD of drugs having both AG and N-AG. Fifty-eight percent of the consumption in DDD of drugs having AG was within the ATC codes of A02BC (proton pump inhibitors— pantoprazole (61.3 million DDD, 312,000 users), lansoprazole (33.6 million DDD, 146,000 users), and omeprazole (37.1 million DDD, 130,480 users)); B01AC (platelet aggregation inhibitors excluding heparin—clopidogrel (36.8 million DDD, 120,000 users)); and C07AB (beta blocking agents— metoprolol (47.7 million DDD, 277,660 users)).

Most of the guidelines in terms of number, both AG and N-AG, exist within the ATC codes N06A (antidepressants drugs), N02A (opioids) including codeine (R05DA04), and N05A (antipsychotic drugs). Due to this, the consumption of drugs within these ATC codes in DDD and number of users who redeemed the drugs were further scrutinized. The total consumption in 2017 were 157.0, 48.9, and 23.7 million DDD for antidepressants, opioids, and antipsychotics, respectively (Figure 1A–C). The consumption of drugs having AG constituted 78.6% and 57.5% of the total consumption in DDD for antidepressants and opioids, which corresponds to the consumption of 11 and 3 different drugs, respectively (see also Table 1 and Table 2). The same numbers for antipsychotic drugs were 17.6% for the consumption of four drugs.

Table 1, Table 2 and Table 3 show the total consumption in DDD of antidepressants, opioids, and antipsychotics in increasing order as well as number of users who redeemed the drugs. Note that the numbers of users for the different drugs shown are not additive, since dispensing to the same users can occur for the different drugs. The Tables also give an overview of drugs having both actionable-, non-actionable- and no-guidelines at all, as well as drugs having FDA approved annotations in their drug labels containing PGx information for CYP2D6 and CYP2C19 drug–gene pairs. Moreover, drugs known to be inhibitors of CYP2D6 or CYP2C19 [30] are also marked. For antidepressants sertraline, citalopram, and venlafaxine (Table 1) both in terms of DDD and number of users were the most redeemed drugs. The consumption of these three drugs constituted approximately 50% of total consumption of DDD in this drug class and they all have AG and, for citalopram and venlafaxine, FDA annotations as well. 

For analgesics (Table 2), the consumption of opioids including codeine in DDD constituted 24.8% of the total consumption of analgesics. Both tramadol and codeine have AG and FDA annotations and the consumption of these drugs in both DDD and number of users constituted more than 50% of total consumption of opioids. Since the content of codeine in N2AJ06 equals that of codeine (R05DA04) i.e., 25 mg, opposite N02AJ07 where the content is three times less, the notion AG and FDA annotation was added to this combination of codeine and paracetamol.

In case of antipsychotic drugs (Table 3) the total consumption in DDD and users is somewhat less compared to opioids and antidepressants and only four drugs out of 24 drugs have AG (Table 3 shows 16 drugs. See legend to Table), however, the highest prevalence of non-AG exists within this drug class. Additionally, in this class of drugs, the annotation “test required” is seen for pimozide. Ariprazole is the most selling drug having both and AG and FDA annotation. For most, but not all, of the drugs shown in the tables there is a good match between those having AG and FDA annotations.

Table 4 shows the number of users who redeemed on the same day or during 2017, a combination of drugs (antidepressants) known as inhibitors of CYP2D6 [30] together with either codeine, tramadol, metoprolol, or amitriptyline. All drugs in the table have AG. From the table it can be seen that e.g., 264,065 users redeemed tramadol, 30,405 users redeemed duloxetine, and 5,544 users redeemed the combination of tramadol and duloxetine during 2017, and 2,126 redeemed the combination on the same day. It can be calculated from the table that users who redeemed the combinations of drugs on the same day as percentage of the whole year ranged from 27.8% for the combination of sertraline and codeine to 63.3% for paroxetine and metoprolol with a median value of 41.7%. The highest prevalence’s for combinations were seen for sertraline in combination with tramadol or metoprolol for both the same day and the whole year. By using the drug interaction tracker provided by Medscape [25] the aim was to visualize warnings “monitor closely” and “serious use alternate” for different drug combinations shown. Most of the drug combinations in the Table 4 were scored by the tracker as either “monitor closely” or “serious use alternate” in relation to CYP2D6. 

Table 5 shows age-distribution expressed as prevalence of users who redeemed tramadol, metoprolol and sertraline either alone or in combination during 2017. The prevalence of tramadol and metoprolol redeemed alone or in combination with sertraline ((S + T) and (S + M)) increased as function of increasing age intervals with the highest prevalence seen for 80+ years. The prevalence of sertraline varies around 18–22 for most age intervals with the highest prevalence seen for 80+ years. The Table also shows (right part) the prevalence of sertraline in combination with tramadol (S + T) or in combination with metoprolol (S + M) expressed relative to the prevalence of sertraline, tramadol and metoprolol. As seen, the relative prevalence of S + T and S + M when expressed relative to sertraline increases as function of age intervals, whereas the relative prevalence of S + T and S + M relative to tramadol and metoprolol did not vary much across age intervals. 

## 4. Discussion

This study visualizes the drug consumption in Denmark in 2017 for drugs having CPIC and/or DPWG dosing guidelines and FDA annotations and suggests future directions, especially with focus on vulnerable polypharmacy patients. Pharmacogenomics based prescribing is one of the first applications of genomics in medicine [21] and testing of polymorphism of genes encoding CYP450 enzymes and in particular CYP2D6 and CYP2C19, but also CYP2C9 has the potential to be a valuable tool to optimize prescribing, dosing, and monitoring by identifying phenotype status [17,18,19,31] and to reduce re-hospitalization in polypharmacy patients [32]. However, significant barriers still exist mainly concerning physicians´ and pharmacists´ awareness and education, and question marks about evidence levels and significance are brought into the discussions. In Denmark the situation does not look much different and the use of PGx tests have not gained foothold in daily clinical practice [22] mainly because of significant organizational barriers and lack of evidence. It has been estimated that around 750,000 Danes are polypharmacy users defined as taking ≥5 drugs corresponding to 13% of the Danish population and for 75+ years, 54% of this age group are polypharmacy users [32]. Taken the average Caucasian frequencies of DGI recently reported [31] for CYP2D6 (UM 4.1%, IM 6.2%, and PM 5.4%) and for CYP2C19 (UM 31.5%, 26.9%, and PM 2.6%) into consideration further suggests that a significant proportion of the Danish population will have phenotypes for which actions in principle should be taken regarding dose adjustments or avoidance of the given drugs. By using the above numbers, which are in alignment with other reported prevalence’s [33], it can roughly be estimated that approximately 0.9 million of the total Danish population of 575 million will be either UM, IM, or PM for CYP2D6 and approximately 3.5 million for CYP2C19, and for polypharmacy users the numbers are approximately 0.1 and 0.5 million, respectively, of which the elderly constitutes 50% (see above). Medication reviews performed in two Danish nursing homes, of which all residents were polypharmacy users, revealed that 80% of the residents were administered drugs for which there exists AG (unpublished results). This underscores the findings of this study showing that a large portion of the Danish population, in particular the elderly, has been or will be exposed to one or more drugs having AG. Citalopram, clopidogrel, lansoprazole, omeprazole, pantoprazole, and sertraline were each redeemed to more than 100,000 users, and metoprolol and tramadol to more than 250,000 users. For example, the dosing guidelines for tramadol (https://www.pharmgkb.org) alert to an action to be taken for CYP2D6 IM, PM, or UM metabolizers, and based on the above frequencies for UM, IM, and PM for CYP2D6 approximately 41,000 of the 265,030 users who redeemed tramadol could potentially be considered for an action to be taken. 

In spite of that a large number of antipsychotic drug are metabolized by CYP2D6 and CYP2C19, only a very few have AG and FDA annotations, with aripiprazol as the most redeemed (11,900 users) drug with an AG. However, due to the complexity of dosing antipsychotic drugs several dosing models based on PGx testing (among others) have been proposed [34,35].

The publically available drug-interaction checker provided by MedScape, used in this study to score for severity of DDI, showed that for the drug combinations investigated, mostly all had warnings related to inhibition of CYP2D6 activity by duloxetine, fluoxetine, paroxetine, and sertraline. The number of users affected by the warning “serious use alternate” seem to be relatively low compared to the number of user affected by the warning “monitor closely”. Surprisingly, the fraction of drug users who redeemed the drug combinations investigated for on the same day seem to be relative high (median value of 41.7%) compared to the whole year. Personal experiences from the pharmacy practice is, that the warning “monitor closely” is very seldom taken into consideration unless the user has undergone a medication review. The warning “serious use alternate”, if caught, is in principle simpler to handle due to the clear message of action. However, independently of the drug combinations are redeemed on the same day or during the same year, actions can only be taken if it is brought to the attention of the pharmacist and physician. This study does not provide data on how many unwanted drug combinations that have been captured during the dispensing process. If the impact of being scored as PM or UM is also taken into consideration, the balance between “monitor closely” and “serious use alternate” might change significantly towards the latter, which recently has been substantiated for opioids [17,36] and for metoprolol [37]. It should be mentioned that other drug-interaction checkers might score the warnings shown in Table 5 differently from that obtained in this study, e.g., “Interaktionsdatabasen” provided by the Danish Medicines Agency (http://www.interaktionsdatabasen.dk/); however, it is the author’s opinion that the principles outlined above still will be valid. 

The increased prevalence’s of tramadol and metoprolol redeemed alone or in combination with sertraline as function of age intervals seen in this study support the notion that elderly people are more exposed to inappropriate polypharmacy than younger people are [38]. The term “used or redeemed alone” in this study does not exclude that users could have redeemed other drug combinations not looked for. The prevalence’s of sertraline in combination with tramadol (S + T) or in combination with metoprolol (S + M) expressed relative to the prevalence of sertraline, tramadol, and metoprolol alone (Table 5) suggests that the fraction of sertraline users who get tramadol or metoprolol increases as a function of age. The opposite is found when expressed as relative to the prevalence of tramadol or metoprolol, as the fraction of metoprolol and tramadol users getting sertraline is constant across age intervals. The same pattern was seen for duloxetine in combination with tramadol and metoprolol (results not shown). In conclusion, the limitations of this register study are that the numbers of users who redeemed the drugs searched for in this publication does not provide any information about dose, compliance, and clinical effects, as well as duration of treatments. However, the findings of this exploratory register study clearly showed that a large fraction of the Danish population, especially the elderly part, are exposed to drugs or drug combinations for which there exist dosing guidelines provided by CPIC or DPWG as well as FDA annotation related to PGx of CYP2D6 and CYP2C19. This underscores the importance of accessing and accounting for DDI, DGI, and DDGI, focusing on the elderly vulnerably (e.g., those in nursing homes) in the first place, while understanding that it is a complex process demanding multidisciplinary collaborations to obtain infrastructural capacities for good decision-making processes, as well as further studies to assess the economic impact of pre-emptive PGx-panel testing.

## Figures and Tables

**Figure 1 jpm-10-00003-f001:**
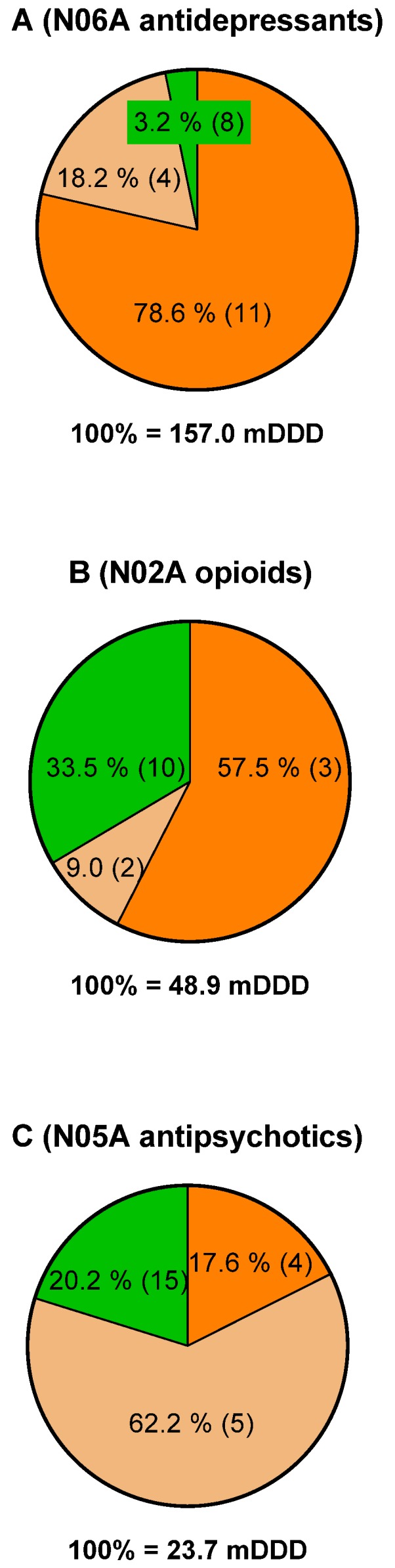
Percentage of total consumption in DDD in 2017 of drugs having actionable guidelines (orange), non-actionable guidelines (light brown), and no guideline (green) related to CYP2D6 and/or CYP2C19 for antidepressants (**A**), opioids (**B**), and antipsychotics (**C**). Numbers in brackets refer to the number of drugs in each category.

**Table 1 jpm-10-00003-t001:** Consumption of antidepressants.

Antidepressants	ATC-Code	CYP2D6	CYP2C19	FDA Ann.	Consumption (Users)	Consumption (DDD) (×1000)
Total antidepressants	N06A				414,850	156,968
Sertraline ^1^	N06AB06	-	AG	-	100,925	43,690 (27.8)
Citalopram	N06AB04	-	AG	A	100,125	34,554 (22.0)
Venlafaxine	N06AX16	AG	-	A	50,155	22,757 (14.5)
Mirtazapine	N06AX11	NG	NG	-	82,300	15,637 (10.0)
Escitalopram	N06AB10	-	AG	A	24,020	9259 (5.9)
Duloxetine ^1^	N06AX21	NG	-	A	30,500	8613 (5.4)
Paroxetine ^1^	N06AB05	AG	-	I	12,935	4960 (3.1)
Fluoxetine ^1,2^	N06AB03	NG	NG	I	10,815	4207 (2.7)
Amitriptyline	N06AA09	AG	AG	A	35,020	3625 (2.3)
Nortriptyline	N06AA10	AG		A	14,965	3222 (2.1)
Agomelatine	N06AX22	-	-		5370	1827 (1.1)
Vortioxetine	N06AX26	-	-	A	5700	1601 (1.0)
Mianserin	N06AX03	-	-	-	12,070	1081 (<1)
Clomipramine ^1^	N06AA04	AG	AG	A	2775	745 (<1)
Imipramine	N06AA02	AG	AG	A	4085	441 (<1)
Isocarboxazid	N06AF01	-	-	-	380	227 (<1)
Bupropion ^1^	N06AX12	-	-	-	2005	149 (<1)
Doxepin ^1^	N06AA12	AG	AG	A	485	99 (<1)
Dosulepin	N06AA16	-	-	-	405	92 (<1)
Reboxetine	N06AX18	-	-		375	79 (<1)
Fluvoxamine ^2^	N06AB08	AG	-	A	185	57 (<1)
Moclobemide	N06AG02	-	NG	-	120	44 (<1)
Maprotiline	N06AA21	-	-	-	20	3 (<1)

The table is sorted in order of increasing consumption. AG: actionable guideline, N-AG: non-actionable guideline, and -: no guideline. FDA annotation (FDA ann.) related to CYP2D6 and/or CYP2C19. Numbers in brackets are percentage expressed relative to the total consumption of antidepressants in DDD: A = action, I = informative; T-req = test required. DDD: Defined Daily Dose (DDD). ^1^ Inhibitor of CYP2D6 and ^2^ inhibitor CYP2C19 [30].

**Table 2 jpm-10-00003-t002:** Consumption of analgesics.

Analgesics	ATC Code	CYP2D6	CYP2C19	FDA Ann.	Consumption (Users)	Consumption (DDD) (×1000)
Total analgesics	N02				1,231,305	196,593
Total opioids	N02A				427,765	44,508 48,875 * (24.8)
Tramadol	N02AX02	AG		A	265,030	20,520 (42.0)
Morphine	N02AA01				97,765	5698 (11.7)
Codeine & acetylsalicylic acid)	N02AJ07				20,980	5536 (11.3)
Oxycodone	N02AA05	NG			69,470	4416 (9.0)
Codeine *	R05DA04	AG		A	71,135	4367 (8.9)
Codeine & paracetamol **	N02AJ06	(AG)		(A)	24,210	3209 (6.6)
Fentanyl	N02AB03				18,815	3046 (6.2)
Buprenorphine	N02AE01				17,470	1132 (2.3)
Ketobemidone and antispasmodics	N02AG02				6085	419 (<1)
Tapentadol	N02AX06				2110	302 (<1)
Hydromorphone	N02AA03				85	124 (<1)

See Table 1 for legend. * Codeine (R05DA04) is included in the total consumption of opioids in DDD (44,508 + 4367 = 48,875 (1000)). Numbers in brackets are percentage expressed relative to total consumption of opioids incl. codeine in DDD. ** Since codeine is approximately 25 mg in N02AJ06, opposite N02AJ07 where the content is three times less, the term AG was applied. Drugs or drug combinations with a consumption of less of 100,000 DDD are not shown.

**Table 3 jpm-10-00003-t003:** Consumption of antipsychotics.

Antipsychotics	ATC Code	CYP2D6	CYP2C19	FDA Ann.	Consumption (Users)	Consumption (DDD) (×1000)
Total antipsychotics	N05A				128,935	23,705
Quetiapine	N05AH04	NG	-	-	61,665	5991 (25.3)
Olanzapine	N05AH03	NG	-	-	17,335	5389 (22.7)
Aripiprazole ^1^	N05AX12	AG	-	A	11,900	2711 (11.7)
Lithium	N05AN01	-	-	-	8925	2175 (9.2)
Risperidone	N05AX08	NG	-	I	16,080	1931 (8.1)
Clozapine	N05AH02	NG	-	A	3375	1118 (4.7)
Zuclopenthixol	N05AF05	AG	-	-	4475	815 (3.4)
Chlorprothixene	N05AF03	-	-	-	15,785	689 (2.9)
Paliperidone	N05AX13	-	-	-	1490	587 (2.5)
Haloperidol ^1^	N05AD01	AG	-	-	6290	532 (2.2)
Ziprasidone	N05AE04	-	-	-	1145	474 (2.0)
Flupenthixol	N05AF01	NG	-	-	4305	308 (1.3)
Perphenazine	N05AB03	-	-	-	575	270 (1.1)
Amisulpride	N05AL05	-	-	-	740	189 (<1)
Levomepromazine	N05AA02	-	-	-	4605	136 (<1)
Pimozide	N05AG02	AG	-	T-req	615	110 (<1)

See Table 1 for legend. Numbers in brackets are percentage expressed relative to total consumption of antipsychotics in DDD. T-req = test required and I = informative. Drugs with a consumption of less of 100,000 DDD are not shown.

**Table 4 jpm-10-00003-t004:** Consumption of drug combinations having Clinical Pharmacogenetics Implementation Consortium CPIC/Dutch Pharmacogenetics Working Group (DPWG) dosing guidelines.

		Codeine	Tramadol	Metoprolol	Amitriptyline
		70,853	264,065	277,090	34,922
Duloxetine ^1^	30,405	1153 (355)	**5544 (2126)**	2596 (1399)	**1825 * (592)**
Fluoxetine ^1^	10,733	347/0.06 (127)	1102 * (443)	**699 (409)**	**158 (77)**
Paroxetine ^1^	12,881	341/0.06 (108)	1202 (441)	**1203 (761)**	**183 * (82)**
Sertraline ^1^	100,490	2644 (734)	11,221 (3962)	8014 (4156)	**1496 * (582)**

Data are presented as total no of users who redeemed the drugs shown either alone (upper and left panel) or in combination during 2017. Numbers in brackets show total number of users who redeemed the drug combinations on the same day. ^1^ Inhibitor of CYP2D6. Drug–drug interactions are scored by using MedScape [25]. Underlined: monitor closely; bold: serious use alternate. All warnings are related to CYP2D6 activity except *. This warning refers to increases in serotonin levels.

**Table 5 jpm-10-00003-t005:** Age distribution of consumption in terms of prevalence.

	Sertraline	Tramadol	Metoprolol	(S + T)	(S + M)	(S + T)/S	(S + T)/T	(S + M)/S	(S + M)/M
Age	Users/1000 Inhabitants					(%)			
00–19	4.0	2.3	0.6	0.1	0.0	1.3	2.3	0.4	2.3
20–29	20.2	18.9	2.3	0.9	0.1	4.5	4.8	0.7	6.1
30–39	23.1	34.1	5.4	1.8	0.3	7.6	5.2	1.4	5.7
40–49	22.3	49.8	18.3	2.4	0.8	10.7	4.8	3.8	4.6
50–59	19.5	62.9	47.7	2.6	1.5	13.3	4.1	7.8	3.2
60–69	18.2	76.4	106.4	2.8	2.7	15.5	3.7	14.6	2.5
70–79	20.6	92.5	161.9	3.7	4.0	18.0	4.0	19.6	2.5
80–	29.3	117.2	205.5	5.1	6.4	17.4	4.3	21.9	3.1
All	17.5	45.9	48.2	2.0	1.4	11.2	4.2	8.0	2.9

Left part of the Table: Age distribution of consumption expressed in terms of prevalence. (S + T): sertraline redeemed in combination with tramadol; (S + M): sertraline redeemed in combination with metoprolol. “All” is the prevalence across all age groups (bottom row). Right part of the table: prevalence in percentage of (S + T) and (S + M) relative to (S) sertraline, (T) tramadol and (M) metoprolol.

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
