# Peer review of "Drug Use in Denmark for Drugs Having Pharmacogenomics (PGx) Based Dosing Guidelines from CPIC or DPWG for CYP2D6 and CYP2C19 Drug–Gene Pairs: Perspectives for Introducing PGx Test to Polypharmacy Patients"

_jpm, 2020, doi:10.3390/jpm10010003_

Round 1

Reviewer 1 Report

The manuscript by Westergaard et al. investigates the use of drugs in Denmark for drugs having actionable pharmacogenomics-based dosing guidelines. The authors analyzed the number of drug users in 2017 for 28 drugs metabolized by CYP2D6 and/or CYP2C19. The study was based on data retrieved from the Danish Register of Medicinal Product Statistics. The results indicate that the large fraction of the Danish population is exposed to drugs for which there exist dosing guidelines related to CYP2D6 and CYP2C19 genotypes. The study presents perspectives for introducing of pharmacogenetics tests in the Danish population. The analysis was well planned and carried out. The results and conclusion are valuable. The space for the implementation of PGx testing was confirmed. However, some additional points would be introduced into the discussion.

The authors should specify and discuss the actual barriers for the full implementation of the testing. The topics would include still limited power to predict the drug effects due to insufficient reporting of drug-genes associations and the high complexity of CYP* loci. Moreover, genome-based poly-drug optimization will require the development of databases and informatics tools to process and interpret the variability in pharmacogenes.

Reviewer 2 Report

Abstract: it would be helpful for the reader if there was a brief description or definition of Defined Daily Dose (i.e. how is this measured in the database and what does it refer to). Additionally, because the reader may not be aware of Denmark demographics, it might be helpful to include some sort of denominator, e.g. total number of drugs prescribed or DDD during the timeframe reviewed. This will provide the reader with an idea of how often drugs with AG are prescribed relative to all prescriptions. Lastly, the authors state “…especially the elderly part,” in the conclusion of the abstract, but there’s no reference to elderly in the results or methods. As mentioned previously, for the results, it would be interesting to learn of all DDDs/prescribed medications in Denmark, what percentage had CPIC/DPWG guidelines? I believe this is shown in Figure I for antidepressants, opioids and antipsychotics, but it would be helpful to have the overall number, i.e. 469.1 DDD with CPIC/DPWG guidelines out of _____ total DDD prescribed (total consumption of all classes of drugs). For tables 1, 2, and 3, it would be helpful to include the (%) for each drug, i.e. percent consumption of sertraline out of total antidepressants. Lines 182-208 are very confusing the way it is currently written. This is not intuitive to readers and is not clear what table 4 is showing. There should be a paragraph break at line 199. The entire section on table 4 and 5 needs to be reconsidered and illustrated in a more simple format, or excluded altogether. To provide some perspective on how many patients in Denmark could have been impacted by receiving a drug with a known gene interaction, consider using population frequency data for % of poor, intermediate, normal, and ultrarapid metabolizers for each CYP2D6 and CYP2C19 (using published data for Europeans or if available for Danish). For example, if the proposed frequency of CYP2D6 poor metabolizers in Denmark is 8%, then one could make the statement that “_____ number of people prescribed opioids may have altered clinical response”. This could be done for the major drug classes for both genes to show how many patient may have altered drug response based on prescribing patterns and population frequency data for genotypes. The discussion hints to some of this, but not with much detail. If this were included in the results it would be much more informative. The discussion should ideally begin with the most important takeaway message from this study – what were the main results and why are they important (instead of discussing a different study first)? Then, lead into other studies and how they support your findings. The title has the word “polypharmacy” in the title, but there is no mention of polypharmacy in the results. Typically, the definition of polypharmacy is the use of 5 or more prescribed medications.

Round 2

Reviewer 2 Report

Please find attached a couple comments

Author Response

To the Editor/reviewer

Thank you very much for the constructive comments to our manuscript.

Attached please find our responses.

We hope that our manuscript is now suitable for publication in JPM

Niels Westergaard
